

# A prediction modeling based on SNOT-22 score for endoscopic nasal septoplasty: a retrospective study

Xue-ran Kang[1,2,3,*], Bin Chen[1,2,3,*], Yi-sheng Chen[4], Bin Yi[1,2,3], Xiaojun Yan[1,2,3], Chenyan Jiang[1,2,3], Shulun Wang[1,2,3], Lixing Lu[1,2,3] and Runjie Shi[1,2,3]

[1] Department of Otorhinolaryngology Head and Neck Surgery, Shanghai ninth people's Hospital, Shanghai Jiao Tong University School of Medicine, Shanghai, China
[2] Ear Institute, Shanghai JiaoTong University School of Medicine, Shanghai, China
[3] Shanghai Key Laboratory of Translational Medicine on Ear and Nose diseases, Shanghai, China
[4] Department of Orthopedics, Shanghai General Hospital, Shanghai Jiao Tong University School of Medicine, Shanghai, China
[*] These authors contributed equally to this work.

## ABSTRACT

**Background.** To create a nomogram prediction model for the efficacy of endoscopic nasal septoplasty, and the likelihood of patient benefiting from the operation.

**Methods.** A retrospective analysis of 155 patients with nasal septum deviation (NSD) was performed to develop a predictive model for the efficacy of endoscopic nasal septoplasty. Quality of life (QoL) data was collected before and after surgery using Sinonasal Outcome Test-22 (SNOT-22) scores to evaluate the surgical outcome. An effective surgical outcome was defined as a SNOT-22 score change $\geq$ 9 points after surgery. Multivariate logistic regression analysis was then used to establish a predictive model for the NSD treatment. The predictive quality and clinical utility of the predictive model were assessed by C-index, calibration plots, and decision curve analysis.

**Results.** The identified risk factors for inclusion in the predictive model were included. The model had a good predictive power, with a AUC of 0.920 in the training group and a C index of 0.911 in the overall sample. Decision curve analysis revealed that the prediction model had a good clinical applicability.

**Conclusions.** Our prediction model is efficient in predicting the efficacy of endoscopic surgery for NSD through evaluation of factors including: history of nasal surgery, preoperative SNOT-22 score, sinusitis, middle turbinate plasty, BMI, smoking, follow-up time, seasonal allergies, and advanced age. Therefore, it can be cost-effective for individualized preoperative assessment.

## INTRODUCTION

Nasal septum deviation (NSD) is one of the most frequently encountered diseases in the rhinology clinic in which the nasal septum deviates from the midline, causing the nasal cavity to shrink in size (*Cho et al., 2010*; *Mattos, Woodard & Payne, 2011*). It occurs in 77–90% of the general population (*Gray, 1978*; *Mladina et al., 2008*). Common symptoms

Corresponding author
Runjie Shi, runjieshi@hotmail.com

are stuffy nose, headache, nosebleed, etc. NSD is an expensive medical condition, and also acts as a predisposing factor for other diseases (*Bousquet et al., 2008*; *Fokkens et al., 2012*). For instance, long-term NSD may indirectly increase the risk of cardiovascular disease (*Ozkececi et al., 2016*; *Uluyol et al., 2016*). Conservative treatments for NSD have a low benefit likelihood (*Lindsay, 2012*; *Rhee et al., 2014*; *Rudy & Most, 2017*), and according to studies published in The Lancet in 2019, nasal septoplasty is more effective than non-surgical treatment in adults with NSD (*Van Egmond et al., 2019*). Therefore, in NSD treatment, surgical method should be considered over conservative non-surgical method (*Aydogdu et al., 2019*). The use of endoscopy provides better illumination for nasal septum correction, making it easier to manage spur or crest. Compared with traditional surgical methods, endoscopic nasal septoplasty surgery has fewer complications and better results. Since it was first described by Stammberger in 1991, it has been increasingly adopted by clinicians and patients (*Dell'Aversana Orabona et al., 2018*). However, the surgical outcome of NSD correction is affected by many factors including: disease-related factors (such as the severity of the disease, combined with sinusitis and middle turbinate plasty); treatment-related factors (such as surgery and rehabilitation protocol, follow-up time, and other medical-related medications); and patient-related factors (such as gender, BMI, age, smoking, drinking history, and seasonal allergies) (*Ahn et al., 2016*; *Alakarppa et al., 2018*; *Becker et al., 2008*; *Hong et al., 2015*). In addition, studies have shown that nasal valve collapse and sinusitis, caused by a history of nasal surgery, could make a profound impact on the efficacy of NSD surgery (*Becker et al., 2008*; *Bhattacharyya, 2005*).

Despite a number of variables that influence the efficacy of NSD surgery being identified in previous studies, no systematic assessment aimed at predicting the surgery's efficacy exists. Therefore, accurate predictive tools and early individualized interventions could be effective in improving the surgical outcomes in patients (*Van Egmond et al., 2015*). SNOT-22 is deemed a credible, valid and responsive disease-specific instrument (*Dietz de Loos et al., 2013*; *Hopkins et al., 2009*; *Morley & Sharp, 2006*). In addition, previous researches support that SNOT-22 is an effective and reliable tool in assessing the results of nasal septal surgery or septorhinoplasty (*Aydogdu et al., 2019*; *Buckland, Thomas & Harries, 2003*; *Poirrier et al., 2013*). Creating a prediction model based on SNOT-22 score for endoscopic nasal septoplasty was the main objective of this study (*Alakarppa et al., 2017*; *Alakarppa et al., 2018*; *Van Egmond et al., 2019*). We hypothesized that based on the degree of improvement in SNOT-22 scores after nasal septoplasty, an effective nomogram model can be developed and used in predicting the likelihood of a benefit outcome from the surgery.

This study sought to develop a simple and effective predictive tool that can be used by clinicians for the efficacy prediction of nasal septoplasty.

## MATERIALS & METHODS

### Research object

This was a retrospective study conducted to establish the efficacy of nasal septoplasty. The study was approved by the Ethics Committee of the Ninth People's Hospital affiliated to

Shanghai Jiao Tong University Medical School (approval no. 2017-323-T243), and met the requirements of the Helsinki Declaration. Eligible patients had all been diagnosed with NSD after an electronic- nasopharyngoscopy and maxillofacial CT scan between January 2015 and September 2019, at the Department of Otorhinolaryngology, Shanghai Ninth People's Hospital. All the patients were residents of China and had undergone an endoscopic nasal septoplasty, conducted by the same surgeon. Combined with preoperative medical information, we conducted further questionnaire surveys and postoperative follow-ups from February to September 2019 by telephone appointment, outpatient and community follow-up. Prior to the study, we obtained signed informed consent forms from the participating patients. Patients enrolled in the study underwent SNOT-22 scores before and after surgery, and all questionnaires and follow-up were successfully completed. In addition, records of basic clinical characteristics such as age, BMI, etc., were taken from the participants. Difference in SNOT-22 score before and after surgery exceeding 9 points is deemed an effective surgical outcome (*Alakarppa et al., 2017*).

## Subject exclusion criteria

All patients attending these clinics completed the Sino Nasal Outcome Test (SNOT-22). The following criteria were set for excluding subjects from the study: Patients who had contracted nasal tumors diseases or nasal septal perforation; lack of cooperation in completing basic diagnostic tests; fever or infection of unknown origin; and acute heart failure, renal failure or any other organ failure. Among the screened patients, a total of 155 patients met the inclusion criteria were included in the study,including 109 males, aged (14-66 years) with a mean age of (35.3 ± 13.5) years, and 46 females, aged (17–62 years) with a mean age of (36.7 ± 12.1) years. We conducted a retrospective analysis on these 155 patients.

## Determination of the curative effect of nasal septoplasty

Sinonasal Outcome Test-22 (SNOT-22) was used to evaluate the disease-specific QoL scores. SNOT-22 is an effective and reliable tool for patients with NSD (*Buckland, Thomas & Harries, 2003*; *Dietz de Loos et al., 2013*; *Hopkins et al., 2009*; *Hytonen et al., 2012*; *Morley & Sharp, 2006*; *Poirrier et al., 2013*). Each question has 0-5 points, and all questions are summed to give a total score between 0 and 110. The higher the SNOT-22 score, the worse the patient's QoL. Postoperative SNOT-22 score change ≥ 9 points (*Alakarppa et al., 2018*; *Hopkins et al., 2009*) was defined as an effective QoL outcome. In this study used the Chinese scale of the SNOT score(*Cao et al., 2017*).

## Medical history and basic data collection

Subjects responded to self-administered questionnaires on the following characteristics: age, gender, smoking, alcohol status, and chronic disease, history of seasonal allergy symptoms, family history, preoperative nasal steroid use history, history of nasal surgery, nasal bone fracture history, follow-up time etc. Medical history data was reviewed to establish whether the subjects had the following medical conditions; nasal polyps, sinusitis, epistaxis, curved/angulated deviation, and spur or crest. Besides the nasal septoplasty, we reviewed whether the subjects had undergone a simultaneous operation such as the middle

turbinoplasty, inferior turbinoplasty, augmentation rhinoplasty, RFVR of the inferior turbinate,nasal bone fracture reduction, endoscopic sinus surgery. And There are many types of morphological classification methods for nasal septal deviation. In this study, we employed the method proposed by Hong-Ryul Jin to classify the nasal septal deviation into four types (*Jin, Lee & Jung, 2007*). The patient's length of hospital stay (day), follow-up period and other data were collected too. The patients were assisted by a physician to complete the SNOT-22 score.

## Statistical analysis

155 selected patients were randomly divided into training and validation (7:3) group for diagnostic and prognostic analysis and used for evaluation of the model. Statistical analysis was performed by R software (version 3.5.3). All pre-determined factors were included in the least absolute shrinkage and selection operator (LASSO) analysis data reduction, and screening of appropriate predictors (*Friedman, Hastie & Tibshirani, 2010*; *Kidd et al., 2018*; *Sauerbrei, Royston & Binder, 2007*). In the Lasso model, a five-fold cross-validation approach was used for the choice of optimal parameters (*Le & Nguyen, 2019*; *Le, Yapp & Yeh, 2019b*). These features were further filtered using a Support Vector Machine-Recursive Feature Elimination (SVM-RFE) algorithm. Support vector machines are increasingly used in the field of bioinformatics (*Le, 2019*; *Le et al., 2019a*). The aim of this method was to find an optimal plane in the multidimensional space, which could divide all sample units into two classes, and also maximize the distance between the two nearest points in the different classes. The edge point between the two closest points is known as the SVM, and the split hyperplane is found in the space between them. The SVM was then used as a classifier for the SVM-RFE algorithm, from the most relevant to the least relevant ordering of the features. The SVM-RFE algorithm may be superior to the linear discriminant analysis, and the mean square error method in selecting relevant features and removing redundant features, especially in the case of a small number of samples (*Huang et al., 2014*).

A predictive model was then built using a multivariate logistic regression model. Based on the collected patient data, we established a predictive model including all the best predictors, to predict the outcome of surgical correction in patients with NSD (*Balachandran et al., 2015*; *Iasonos et al., 2008*). We plotted the calibration curve to evaluate the nomogram's accuracy (*Kramer & Zimmerman, 2007*). To further quantify the discrimination performance of the nomogram, we measured the c-index and the AUC. R language package was then used to perform further iterations (10,000 repetitive samples) on the nomogram to calculate a more accurate C-index (*Pencina & D'Agostino, 2004*). In addition, we also used an external dataset to corroborate our results. By quantifying the net benefit of different threshold probabilities in patient information, decision curve analysis was used to assess the clinical utility of the nomogram. All statistical tests were bilateral and statistical significance was set at $P < 0.05$. This study was conducted in line with the Transparent Reporting of a Multivariate Prediction Model for Individual Prediction or Diagnosis guidelines (*Collins et al., 2015*).

## RESULTS

### Basic characteristics of data

We investigated postoperative information from 155 patients with NSD (109 males and 46 females) from January 2015 to April 2019. The patients were divided into an improved group (127 patients) and unimproved group (28 patients) according to the changes in SNOT-22 scores before and after surgery. Table 1 outlines the overall characteristics of the patients data.

### Screening prediction

We applied the LASSO regression model (Figs. 1A, 1B) to reduce the factors investigated in this study from 48 to 20. On the basis of Harrell's guidelines, when the outcome is binary, the minimum value of the frequencies of the two response levels should be greater than 10 times the number of predictors (*Harrell, 2001*; *Iasonos et al., 2008*). As shown in Figs. 1A and 1B in the article, simple use of lasso analysis requires the inclusion of 20 variables in the model. If some variables are not filtered and excluded, it will require that at least 200 samples be included for training cohort, which is difficult to achieve. Nevertheless, we have collected all possible data and added new external data. To address this problem, we screened the factors further using the SVM-RFE algorithm to obtain "the least characteristic factor" (Table 2). The SVM-RFE algorithm may be superior to the linear discriminant analysis and the mean square error method in selecting relevant features and excluding redundant features, especially in the case of a small number of samples (*Huang et al., 2014*). Finally, it proves that the prediction model established by the characteristic factors screened by svm has higher accuracy. The best prediction model had nine optimal features with an average cross-validation score of 0.8681 (Fig. 1C). These characteristics included history of nasal surgery, preoperative SNOT-22 score, sinusitis, middle turbinate plasty, BMI, smoking, follow-up, advanced age, and seasonal allergies. In the future, we will expand the sample size to confirm the performance of the nomogram in predicting the efficacy and clinical utility of NSD correction. As the sample size increases, the composition of optimal characteristic factors may also change, and at that time we will have the conditions necessary to further evaluate which method is more effective in establishing a prediction model.

### Building a personalized predictive model

After logistic model analysis of the nine predictors (Table 3), we used R software nomogramEx package to construct the nomogram, as shown in Fig. 2. Among them, age, SNOT-22 score, history of sinusitis and nasal surgery were revealed as significant factors affecting the efficacy of nasal septum surgery ($P < 0.05$).

### Inspection of the nomogram prediction model

The calibration curve of nomogram was consistent (Fig. 3A), indicating that the model was competent to predict the surgical outcome of NSD. The model had a good predictive power, with a C-index of 0.920 (95% CI [0.854–0.986]), 0.920 (95% CI [0.854–0.986]), 0.834 (95% CI [0.655–1.000]) and 0.765 (95% CI [0.555–0.974]) in the training queue, the

Kang et al. (2020), *PeerJ*, DOI 10.7717/peerj.9890

**Table 1   Differences of demographic and clinical characteristics between effective and ineffective groups.**

| Characteristics | Entire cohort | | | | Training set | | | | Validation set | | | | External dataset | | |
|---|---|---|---|---|---|---|---|---|---|---|---|---|---|---|---|
| | Effective [N = 127] | Ineffective [N = 28] | Total [N = 155] | P-value | Effective (N = 90) | Ineffective (N = 21) | Total (N = 111) | P-value | Effective (N = 37) | Ineffective (N = 7) | Total (N = 44) | P-value | Effective (N = 17) | Ineffective (N = 4) | Total (N = 21) |
| **Gender** | | | | $P < 0.001$ | | | | $P < 0.01$ | | | | $P < 0.01$ | | | |
| Female | 38[30%] | 8[28.6%] | 46[29.7%] | | 57[63.3%] | 16[76.2%] | 73[65.8%] | | 32[86.5%] | 4[57.1%] | 36[81.8%] | | 7(59%) | 1(25%) | 8(38%) |
| Male | 89[70%] | 20[71.4%] | 109[70.3%] | | 33[36.7%] | 5[23.8%] | 38[34.2%] | | 5[13.5%] | 3[42.9%] | 8[18.2%] | | 10(41%) | 3(75%) | 13(62%) |
| **Age** | | | | | | | | | | | | | | | |
| Mean [SD] | 33.9[12.6] | 43.8[12.6] | 35.7[13.1] | | 34.8[12.9] | 43.5[11.8] | 36.5[13.1] | | 31.8[11.7] | 44.9[15.6] | 33.8[13.1] | | 38.6(13.3) | 42.0(22.0) | 39.2(14.7) |
| Median [MIN, MAX] | 31 [14,66] | 45[22,63] | 33[14,66] | | 32.5[14,66] | 47[22,61] | 34[14,66] | | 29[16,64] | 43[22,63] | 30.5[16,64] | | 40 [16,57] | 31.5[30,75] | 39[16,75] |
| **BMI** | | | | $P < 0.01$ | | | | $P < 0.05$ | | | | $P < 0.05$ | | | |
| Mean [SD] | 23.0[2.9] | 24.7[3.2] | 23.3[3.1] | | 22.9[3.6] | 24.8[3.5] | 23.3[3.6] | | 22.7[2.6] | 24.5[2.7] | 23[2.6] | | 22.6(4.4) | 22.8(0.5) | 22.6(3.9) |
| Median [MIN, MAX] | 22.9[16.2,30.9] | 24.3[18.5,31.5] | 23.0[16.2,31.5] | | 23[10.9,29.3] | 24.3[18.5,31.5] | 23.4[10.9,31.5] | | 22.6[17,30.9] | 24.2[21.5,29.8] | 22.7[17,30.9] | | 21.9[17.6,30.9] | 23.0[22.0,23.0] | 22.0[17.6,30.9] |
| **Atrophic Rhinitis** | | | | | | | | | | | | | | | |
| Yes | 1[0.8%] | 3[10.7%] | 4[2.6%] | | 1[1.1%] | 2[9.5%] | 3[2.7%] | | 0[0%] | 1[14.3%] | 1[2.3%] | | 0(0%) | 0(0%) | 0(0%) |
| No | 126[99.2%] | 25[89.3%] | 151[97.4%] | | 89[98.9%] | 19[90.5%] | 108[97.3%] | | 37[100%] | 6[85.7%] | 43[97.7%] | | 17(100%) | 4(100%) | 21(100%) |
| **Nasal Polyps** | | | | | | | | | | | | | | | |
| Yes | 10[7.9%] | 6[21.4%] | 16[10.3%] | | 5[5.6%] | 6[28.6%] | 11[9.9%] | | 5[13.5%] | 0[0%] | 5[11.4%] | | 2(11.8%) | 0(0%) | 2(9.5%) |
| No | 117[92.1%] | 22[78.6%] | 139[89.7%] | | 85[94.4%] | 15[71.4%] | 100[90.1%] | | 32[86.5%] | 7[100%] | 39[88.6%] | | 15(88.2%) | 4(100%) | 19(90.5%) |
| **Epistaxis** | | | | | | | | | | | | | | | |
| Yes | 19[15.0%] | 4[14.3%] | 23[14.8%] | | 16[17.8%] | 4[19%] | 20[18%] | | 3[8.1%] | 0[0%] | 3[6.8%] | | 5(29.4%) | 0(0%) | 5(23.8%) |
| No | 108[85.0%] | 24[85.7%] | 132[85.2%] | | 74[82.2%] | 17[81%] | 91[82%] | | 34[91.9%] | 7[100%] | 41[93.2%] | | 12(70.6%) | 4(100%) | 16(76.2%) |
| **Nasosinusitis** | | | | $P < 0.05$ | | | | | | | | | | | |
| Yes | 23[18.1%] | 16[57.1%] | 39[25.2%] | | 15[16.7%] | 12[57.1%] | 27[24.3%] | | 3(17.6%) | 3(75%) | 6(28.6%) | | 3(17.6%) | 3(75%) | 6(28.6%) |
| No | 104[81.9%] | 12[42.9%] | 116[74.8%] | | 75[83.3%] | 9[42.9%] | 84[75.7%] | | 14(82.4%) | 1(25%) | 15(71.4%) | | 14(82.4%) | 1(25%) | 15(71.4%) |
| **High Blood Pressure** | | | | | | | | | | | | | | | |
| Yes | 18[14.2%] | 7[25.0%] | 25[16.1%] | | 11[12.2%] | 5[23.8%] | 16[14.4%] | | 7[18.9%] | 2[28.6%] | 9[20.5%] | | 4(24%) | 1(25.0%) | 5(24%) |
| No | 109[85.8%] | 21[75.0%] | 130[83.9%] | | 79[87.8%] | 16[76.2%] | 95[85.6%] | | 30[81.1%] | 5[71.4%] | 35[79.5%] | | 13(76%) | 3(75.0%) | 16(76%) |
| **Diabetes** | | | | | | | | | | | | | | | |
| Yes | 7[5.5%] | 3[10.7%] | 10[6.5%] | | 6[6.7%] | 2[9.5%] | 8[7.2%] | | 1[2.7%] | 1[14.3%] | 2[4.5%] | | 3(17.6%) | 0(0%) | 3(14.3%) |
| No | 120[94.5%] | 25[89.3%] | 145[93.5%] | | 84[93.3%] | 19[90.5%] | 103[92.8%] | | 36[97.3%] | 6[85.7%] | 42[95.5%] | | 14(82.4%) | 4(100%) | 18(85.7%) |

*(continued on next page)*
**Table 1** (*continued*)

| Characteristics | Entire cohort | | | | Training set | | | | Validation set | | | | External dataset | | |
|---|---|---|---|---|---|---|---|---|---|---|---|---|---|---|---|
| | Effective [N = 127] | Ineffective [N = 28] | Total [N = 155] | P-value | Effective (N = 90) | Ineffective (N = 21) | Total (N = 111) | P-value | Effective (N = 37) | Ineffective (N = 7) | Total (N = 44) | P-value | Effective (N = 17) | Ineffective (N = 4) | Total (N = 21) |
| **Seansonal Allergy History** | | | | | | | | | | | | | | | |
| Yes | 46[36.2%] | 2[7.1%] | 48[31.0%] | | 34[37.8%] | 1[4.8%] | 35[31.5%] | | 12[32.4%] | 1[14.3%] | 13[29.5%] | | 7[41.2%] | 0(0%) | 7[33.3%] |
| No | 81[63.8%] | 26[92.9%] | 107[69.0%] | | 56[62.2%] | 20[95.2%] | 76[68.5%] | | 25[67.6%] | 6[85.7%] | 31[70.5%] | | 10[58.8%] | 4(100%) | 14[66.7%] |
| **Intranasal Corticosteroids History** | | | | | | | | | | | | | | | |
| Yes | 40[31.5%] | 7[25.0%] | 47[30.3%] | | 29[32.2%] | 5[23.8%] | 34[30.6%] | | 11[29.7%] | 2[28.6%] | 13[29.5%] | | 7[41.2%] | 1(25%) | 8[38.1%] |
| No | 87[68.5%] | 21[75.0%] | 108[69.7%] | | 61[67.8%] | 16[76.2%] | 77[69.4%] | | 26[70.3%] | 5[71.4%] | 31[70.5%] | | 10[58.8%] | 3(75%) | 13[61.9%] |
| **History Of Nasal Bone Fracture** | | | | | | | | | | | | | | | |
| Yes | 10[7.9%] | 5[17.9%] | 15[9.7%] | | 5[5.6%] | 5[23.8%] | 10[9%] | | 5[13.5%] | 0[0%] | 5[11.4%] | | 2(11.8%) | 0(0%) | 2(9.5%) |
| No | 117[92.1%] | 23[82.1%] | 140[90.3%] | | 85[94.4%] | 16[76.2%] | 101[91%] | | 32[86.5%] | 7[100%] | 39[88.6] | | 15(88.2%) | 4(100%) | 19(90.5%) |
| **Nasal Surgery History** | | | P < 0.05 | | | | | | | | | | | | |
| Yes | 13[10.2%] | 12[42.9%] | 25[16.1%] | | 10[11.1%] | 10[47.6%] | 20[18%] | | 8[21.6%] | 4[57.1%] | 12[27.3%] | | 3(17.6%) | 1(25%) | 4(19.0%) |
| No | 114[89.8%] | 16[57.1%] | 130[83.9%] | | 80[88.9%] | 11[52.4%] | 91[82%] | | 29[78.4%] | 3[42.9%] | 32[72.7%] | | 14(82.4%) | 3(75%) | 17(81.0%) |
| **Family History** | | | | | | | | | | | | | | | |
| Yes | 33[26.0%] | 6[21.4%] | 39[25.2%] | | 21[23.3%] | 4[19%] | 25[22.5%] | | 12[32.4%] | 2[28.6%] | 14[31.8%] | | 5(29.4%) | 1(25%) | 6(28.6%) |
| No | 94[74.0%] | 22[78.6%] | 116[74.8%] | | 69[76.7%] | 17[81%] | 86[77.5%] | | 25[67.6%] | 5[71.4%] | 30[68.2%] | | 12(70.6%) | 3(75%) | 15(71.4%) |
| **Smoking** | | | | | | | | | | | | | | | |
| Yes | 31[24.4%] | 10[35.7%] | 41[26.5%] | | 21[23.3%] | 8[38.1%] | 29[26.1%] | | 10[27%] | 2[28.6%] | 12[27.3%] | | 9(52.9) | 2(50.0%) | 11(52.4%) |
| No | 96[75.6%] | 18[64.3%] | 114[73.5%] | | 69[76.7%] | 13[61.9%] | 82[73.9%] | | 27[73%] | 5[71.4%] | 32[72.7%] | | 8(47.1%) | 2(50.0%) | 10(47.6%) |
| **Follow-Up Time[Year]** | | | | | | | | | | | | | | | |
| Mean [SD] | 1.9[1.4] | 2.3[1.3] | 2.0[1.4] | | 2[1.3] | 2.4[1.3] | 2[1.4] | | 1.9[1.4] | 2.1[1.2] | 1.9[1.3] | | 0.38(0.21) | 0.3(0.24) | 0.37(0.21) |
| Median [MIN, MAX] | 1.72[0.03,4.62] | 1.7[0.28,4.43] | 1.7[0.03,4.63] | | 1.8[0,4.3] | 2.8[0.5,4.4] | 1.7[0.03,4.6] | | 1.6[0.1,4.4] | 2.2[0.3,3.4] | 1.8[0.1,4.4] | | 0.43[0.05,0.66] | 0.32[0.06,0.52] | 0.43[0.05,0.66] |
| **Length Of Hospital Stay[Day]** | | | | | | | | | | | | | | | |
| Mean [SD] | 4.8[1.0] | 4.64[0.87] | 4.7[1.0] | | 4.8[1.1] | 4.8[0.9] | 4.8[1] | | 4.6[1.2] | 4.1[0.7] | 4.5[1.1] | | 4.5(0.5) | 4.8(0.5) | 4.5(0.5) |
| Median [MIN, MAX] | 5[3,10] | 5[3,7] | 5[3,10] | | 5[3,10] | 5[4,7] | 5[3,10] | | 4[0,7] | 4[3,5] | 4[0,7] | | 4[4,5] | 5[4,5] | 5[4,5] |
| **Proposed classification system of SD** | | | | | | | | | | | | | | | |
| Type I | 20[15.7%] | 1[3.6%] | 21[13.5%] | | 12[13.3%] | 1[4.8%] | 13[11.7%] | | 8[21.6%] | 0[0%] | 8[18.2%] | | 2[11.8%] | 0[0%] | 2[9.5%] |
| Type II | 83[65.4%] | 22[78.6%] | 105[67.7%] | | 58[64.4%] | 16[76.2%] | 74[66.7%] | | 25[67.6%] | 6[85.7%] | 31[70.5%] | | 1[5.9%] | 2[50%] | 3[14.3%] |
| Type III | 20[15.7%] | 4[14.3%] | 24[15.5%] | | 18[20%] | 3[14.3%] | 21[18.9%] | | 2[5.4%] | 1[14.3%] | 3[6.8%] | | 9[52.9%] | 1[25%] | 10[47.6%] |
| Type IV | 4[3.1%] | 1[3.6%] | 5[3.2%] | | 2[2.2%] | 1[4.8%] | 3[2.7%] | | 2[5.4%] | 0(0%) | 2[4.5%] | | 5[29.4%] | 1[25%] | 6[28.6%] |

Kang et al. (2020), *PeerJ*, DOI 10.7717/peerj.9890

**Table 1** (*continued*)

| Characteristics | Entire cohort | | | | Training set | | | | Validation set | | | | External dataset | | |
|---|---|---|---|---|---|---|---|---|---|---|---|---|---|---|---|
| | Effective [N = 127] | Ineffective [N = 28] | Total [N = 155] | P-value | Effective (N = 90) | Ineffective (N = 21) | Total (N = 111) | P-value | Effective (N = 37) | Ineffective (N = 7) | Total (N = 44) | P-value | Effective (N = 17) | Ineffective (N = 4) | Total (N = 21) |
| **Additional Surgery** | | | | | | | | | | | | | | | |
| **Middle Turbinoplasty** | | | | | | | | | | | | | | | |
| Yes | 11[8.7%] | 2[7.1%] | 13[8.4%] | | 9[10%] | 1[4.8%] | 10[9%] | | 2[5.4%] | 1[14.3%] | 3[6.8%] | | 2(11.8%) | 0(0%) | 2(9.5%) |
| No | 116[91.3%] | 26[92.9%] | 142[91.6%] | | 81[90%] | 20[95.2%] | 101[91%] | | 35[94.6%] | 6[85.7%] | 41[93.2%] | | 15(88.2%) | 4(100%) | 19(90.5%) |
| **Inferior Turbinoplasty** | | | | | | | | | | | | | | | |
| Yes | 58[45.7%] | 10[35.7%] | 68[43.9%] | | 40[44.4%] | 8[38.1%] | 48[43.2%] | | 18[48.6%] | 2[28.6%] | 20[45.5%] | | 7(41.2%) | 4(100%) | 11(52.4%) |
| No | 69[54.3%] | 18[64.3%] | 87[56.1%] | | 50[55.6%] | 13[61.9%] | 63[56.8%] | | 19[51.4%] | 5[71.4%] | 24[54.5%] | | 10(58.8%) | 0(0%) | 10(47.6%) |
| **Augmentation Rhinoplasty** | | | | | | | | | | | | | | | |
| Yes | 8[6.2%] | 0[0%] | 8[5.2%] | | 6[6.7%] | 0[0%] | 6[5.4%] | | 2[5.4%] | 0[0%] | 2[4.5%] | | 3(17.6%) | 0(0%) | 3(14.3%) |
| No | 119[85.8%] | 28[100%] | 147[94.8%] | | 84[93.3%] | 21[100%] | 105[94.6%] | | 35[94.6%] | 7[100%] | 42[95.5%] | | 14(82.4%) | 4(100%) | 18(85.7%) |
| **RFVR Of The Inferior Turbinate** | | | | | | | | | | | | | | | |
| Yes | 8[6.2%] | 1[3.6%] | 9[5.8%] | | 5[5.6%] | 0[0%] | 5[4.5%] | | 3[8.1%] | 1[14.3%] | 4[9.1%] | | 1(5.9%) | 1(25%) | 2(9.5%) |
| No | 109[85.8%] | 27[96.4%] | 146[94.2%] | | 85[94.4%] | 21[100%] | 106[95.5%] | | 34[91.9%] | 6[85.7%] | 40[90.9%] | | 16(94.1%) | 3(75%) | 19(90.5%) |
| **Nasal Bone Fracture Reduction** | | | | | | | | | | | | | | | |
| Yes | 4[3.1%] | 1[3.6%] | 5[3.2%] | | 2[2.2%] | 1[4.8%] | 3[2.7%] | | 2[5.4%] | 0[0%] | 2[4.5%] | | 3(17.6%) | 0(0%) | 3(14.3%) |
| No | 123[96.9%] | 27[96.4%] | 150[96.8%] | | 88[97.8%] | 20[95.2%] | 108[97.3%] | | 35[94.6%] | 7[100%] | 42[95.5%] | | 14(82.4%) | 4(100%) | 18(85.7%) |
| **Endoscopic Sinus Surgery** | | | | | | | | | | | | | | | |
| Yes | 7[5.5%] | 7[25.0%] | 14[9.0%] | | 4[4.4%] | 5[23.8%] | 9[8.1%] | | 3[8.1%] | 2[28.6%] | 5[11.4%] | | 0(0%) | 2(50.0%) | 2(9.5%) |
| No | 120[94.5%] | 21[75.0%] | 141[91.0%] | | 86[95.6%] | 16[76.2%] | 102[91.9%] | | 34[91.9%] | 5[71.4%] | 39[88.6%] | | 17(100%) | 2(50.0%) | 19(90.5%) |
| **SNOT-22** | | | | | | | | | | | | | | | |
| **Need To Blow Nose** | | | | P < 0.001 | | | | P < 0.001 | | | | P < 0.001 | | | |
| Mean [SD] | 3.6[1.0] | 2.9[1.3] | 3.5[1.1] | | 3.7[1] | 2.8[1.3] | 3.5[1.1] | | 3.5[1] | 3.3[1.4] | 3.5[1] | | 3.2[1] | 3.5[1] | 3.3[1] |
| Median [MIN, MAX] | 4.0 [0.0,5.0] | 4.0 [0.0,5.0] | 4.0 [0.0,5.0] | | 4[1,5] | 3[0,4] | 4[0,5] | | 4[1,5] | 4[1,5] | 4[1,5] | | 3[2,5] | 4[2,4] | 3[2,5] |
| **Runny Nose** | | | | P < 0.001 | | | | P < 0.001 | | | | P < 0.01 | | | |
| Mean [SD] | 3.2[1.4] | 2.4[1.8] | 3.1[1.5] | | 3.3[1.4] | 2.3[1.8] | 3.1[1.5] | | 3.1[1.5] | 2.9[1.5] | 3.1[1.5] | | 2.9[1.2] | 2.8[0.5] | 2.9[1.1] |
| Median [MIN, MAX] | 3.0 [0.0,5.0] | 4.0 [0.0,5.0] | 3.0 [0.0,5.0] | | 4[0,5] | 2[0,5] | 3[0,5] | | 3[0,5] | 2[2,5] | 3[0,5] | | 3[1,5] | 3[2,3] | 3[1,5] |
| **Post Nasal Discharge** | | | | P < 0.001 | | | | P < 0.001 | | | | P < 0.01 | | | |
| Mean [SD] | 2.9[1.0] | 2.4[1.4] | 2.8[1.1] | | 2.9[1] | 2.2[1.4] | 2.8[1.1] | | 2.8[1] | 2.9[1.5] | 2.8[1.1] | | 3[0.9] | 4.3[1.5] | 3.2[1.1] |
| Median [MIN, MAX] | 3.0 [0.0,5.0] | 3.0 [0.0,5.0] | 3.0 [0.0,5.0] | | 3[0,5] | 2[0,5] | 3[0,5] | | 3[1,4] | 2[1,5] | 3[1,5] | | 3[2,4] | 5[2,5] | 3[2,5] |
| **Thick Nasal Discharge** | | | | | | | | | | | | | | | |
| Mean [SD] | 2.3[1.2] | 2.2[1.4] | 2.3[1.3] | | 2.4[1.2] | 2.1[1.4] | 2.3[1.3] | | 2.2[1.2] | 2.6[1.5] | 2.3[1.2] | | 2.4[0.6] | 3.8[0.5] | 2.6[0.8] |
| Median [MIN, MAX] | 2.0 [0.0,5.0] | 2.0 [0.0,5.0] | 2.0 [0.0,5.0] | | 2[0,5] | 2[0,5] | 2[0,5] | | 2[0,5] | 2[1,5] | 2[0,5] | | 2[2,4] | 4[3,4] | 2[2,4] |

**Table 1** (*continued*)

| Characteristics | Entire cohort | | | | Training set | | | | Validation set | | | | External dataset | | |
|---|---|---|---|---|---|---|---|---|---|---|---|---|---|---|---|
| | Effective [N = 127] | Ineffective [N = 28] | Total [N = 155] | P-value | Effective (N = 90) | Ineffective (N = 21) | Total (N = 111) | P-value | Effective (N = 37) | Ineffective (N = 7) | Total (N = 44) | P-value | Effective (N = 17) | Ineffective (N = 4) | Total (N = 21) |
| **Sneezing** | | | | $P < 0.001$ | | | | $P < 0.001$ | | | | $P < 0.001$ | | | |
| Mean [SD] | 3.5[1.5] | 2.1[1.5] | 3.2[1.6] | | 3.5[1.5] | 2[1.6] | 3.2[1.6] | | 3.5[1.6] | 2.4[1.3] | 3.3[1.6] | | 3.2[1.5] | 2.5[0.6] | 3[1.4] |
| Median [MIN, MAX] | 4.0 [0.0,5.0] | 4.0 [0.0,5.0] | 4.0 [0.0,5.0] | | 4[0,5] | 2[0,5] | 4[0,5] | | 4[1,5] | 2[1,5] | 3.5[1,5] | | 3[0,5] | 2.5[2,3] | 3[0,5] |
| **Cough** | | | | $P < 0.001$ | | | | $P < 0.001$ | | | | $P < 0.001$ | | | |
| Mean [SD] | 2.5[0.9] | 1.7[1.0] | 2.3[1.0] | | 2.5[0.9] | 1.6[1] | 2.3[1] | | 2.5[0.9] | 2[1] | 2.4[0.9] | | 2.8[0.9] | 2.3[1] | 2.7[0.9] |
| Median [MIN, MAX] | 3.0 [0.0,4.0] | 3.0 [0.0,4.0] | 2.0 [0.0,4.0] | | 3[0,4] | 2[0,4] | 2[0,4] | | 2[1,4] | 2[0,3] | 2[0,4] | | 3[2,5] | 2.5[1,3] | 3[1,5] |
| **Sense Of Taste Or Smell** | | | | | | | | | | | | | | | |
| Mean [SD] | 2.9[1.4] | 2.5[1.3] | 2.9[1.4] | | 3[1.4] | 2.6[1.3] | 2.9[1.4] | | 2.8[1.3] | 2.4[1.5] | 2.7[1.3] | | 2.8[1] | 2.8[0.5] | 2.8[0.9] |
| Median [MIN, MAX] | 3.0 [0.0,5.0] | 3.0 [0.0,5.0] | 3.0 [0.0,5.0] | | 3[0,5] | 3[0,5] | 3[0,5] | | 3[0,5] | 2[0,5] | 3[0,5] | | 3[1,5] | 3[2,3] | 3[1,5] |
| **Blockage/Congestion Of Nose** | | | | $P < 0.001$ | | | | $P < 0.05$ | | | | $P < 0.05$ | | | |
| Mean [SD] | 4.6[1.0] | 3.9[1.6] | 4.4[1.2] | | 4.6[0.9] | 4.1[1.4] | 4.5[1.1] | | 4.5[1.1] | 3.1[1.9] | 4.3[1.3] | | 4.2[1.1] | 4.8[0.5] | 4.3[1.1] |
| Median [MIN, MAX] | 5.0 [0.0,5.0] | 5.0 [0.0,5.0] | 5.0 [0.0,5.0] | | 5[1,5] | 5[0,5] | 5[0,5] | | 5[1,5] | 4[0,5] | 5[0,5] | | 5[1,5] | 5[4,5] | 5[1,5] |
| **Ear Fullness** | | | | | | | | | | | | | | | |
| Mean [SD] | 3.2[1.2] | 2.6[1.3] | 3.1[1.2] | | 3.3[1.2] | 2.8[1.3] | 3.2[1.2] | | 3.2[1.2] | 2.1[1.2] | 3[1.2] | | 3[0.9] | 3.5[1] | 3.1[0.9] |
| Median [MIN, MAX] | 3.0 [0.0,5.0] | 3.0 [0.0,4.0] | 3.0 [0.0,5.0] | | 3[0,5] | 3[0,4] | 3[0,5] | | 3[0,5] | 2[0,4] | 3[0,5] | | 3[1,4] | 4[2,4] | 3[1,4] |
| **Ear Pain** | | | | $P < 0.001$ | | | | $P < 0.001$ | | | | $P < 0.001$ | | | |
| Mean [SD] | 2.5[1.3] | 1.6[1.3] | 2.3[1.3] | | 2.5[1.2] | 1.4[1.2] | 2.3[1.3] | | 2.5[1.4] | 2.3[1.3] | 2.5[1.4] | | 2.2[1.3] | 3.8[0.5] | 2.5[1.3] |
| Median [MIN, MAX] | 2.0 [0.0,5.0] | 2.0 [0.0,4.0] | 2.0 [0.0,5.0] | | 2[0,5] | 1[0,4] | 2[0,5] | | 2[0,5] | 2[0,4] | 2[0,5] | | 2[0,4] | 4[3,4] | 3[0,4] |
| **Dizziness** | | | | $P < 0.001$ | | | | $P < 0.01$ | | | | $P < 0.01$ | | | |
| Mean [SD] | 3.1[1.1] | 2.5[1.5] | 3.0[1.2] | | 3.2[0.9] | 2.4[1.5] | 3[1.1] | | 3.2[1.2] | 2.9[1.5] | 3.1[1.2] | | 2.6[1.1] | 4[0] | 2.9[1.2] |
| Median [MIN, MAX] | 3.0 [0.0,5.0] | 3.0 [0.0,4.0] | 3.0 [0.0,5.0] | | 3[0,5] | 3[0,4] | 3[0,5] | | 3[0,5] | 3[0,4] | 3[0,5] | | 3[0,4] | 4[4,4] | 3[0,4] |
| **Facial Pain** | | | | | | | | | | | | | | | |
| Mean [SD] | 2.9[1.2] | 2.8[1.6] | 2.9[1.2] | | 3[1.1] | 2.7[1.5] | 3[1.2] | | 2.8[1.3] | 3.3[1.8] | 2.9[1.4] | | 2.3[1.1] | 3.5[1] | 2.5[1.2] |
| Median [MIN, MAX] | 3.0 [1.0,5.0] | 3.0 [0.0,5.0] | 3.0 [0.0,5.0] | | 3[1,5] | 3[0,5] | 3[0,5] | | 3[1,5] | 4[0,5] | 3[0,5] | | 2[1,4] | 3[3,5] | 3[1,5] |
| **Difficulty Falling Asleep** | | | | | | | | | | | | | | | |
| Mean [SD] | 2.7[1.2] | 2.6[1.4] | 2.7[1.2] | | 2.7[1.1] | 2.5[1.4] | 2.7[1.1] | | 2.6[1.3] | 3[1.6] | 2.7[1.3] | | 2.3[0.9] | 2.5[1] | 2.3[0.9] |
| Median [MIN, MAX] | 3.0 [0.0,5.0] | 3.0 [0.0,5.0] | 3.0 [0.0,5.0] | | 3[1,5] | 3[0,5] | 3[0,5] | | 3[0,5] | 3[0,5] | 3[0,5] | | 2[1,4] | 2[2,4] | 2[1,4] |
| **Waking Up At Night** | | | | | | | | | | | | | | | |
| Mean [SD] | 2.6[1.2] | 2.5[1.5] | 2.5[1.2] | | 2.6[1.1] | 2.5[1.4] | 2.5[1.2] | | 2.5[1.3] | 2.7[1.8] | 2.6[1.4] | | 2.1[1] | 2[0] | 2[0.9] |
| Median [MIN, MAX] | 3.0 [0.0,5.0] | 3.0 [0.0,5.0] | 3.0 [0.0,5.0] | | 2.5[0,5] | 3[0,5] | 3[0,5] | | 3[0,5] | 2[0,5] | 2.5[0,5] | | 2[1,4] | 2[2,2] | 2[1,4] |
| **Lack Of A Good Sleep** | | | | | | | | | | | | | | | |
| Mean [SD] | 2.7[1.2] | 2.6[1.4] | 2.7[1.2] | | 2.7[1.1] | 2.6[1.5] | 2.7[1.2] | | 2.6[1.2] | 2.9[1.1] | 2.6[1.2] | | 2.3[1.1] | 2.3[0.5] | 2.3[1] |
| Median [MIN, MAX] | 3.0 [0.0,5.0] | 3.0 [0.0,5.0] | 3.0 [0.0,5.0] | | 3[0,5] | 2[0,5] | 3[0,5] | | 3[0,5] | 3[1,4] | 3[0,5] | | 2[1,4] | 2[2,3] | 2[1,4] |

Peerj

**Table 1** (*continued*)

| Characteristics | Entire cohort | | | | Training set | | | | Validation set | | | | External dataset | | |
|---|---|---|---|---|---|---|---|---|---|---|---|---|---|---|---|
| | Effective [N = 127] | Ineffective [N = 28] | Total [N = 155] | *P*-value | Effective (N = 90) | Ineffective (N = 21) | Total (N = 111) | *P*-value | Effective (N = 37) | Ineffective (N = 7) | Total (N = 44) | *P*-value | Effective (N = 17) | Ineffective (N = 4) | Total (N = 21) |
| **Waking Up Tired** | | | | | | | | | | | | | | | |
| Mean [SD] | 2.9[1.1] | 2.6[1.3] | 2.8[1.1] | | 3[1] | 2.5[1.4] | 2.9[1.1] | | 2.8[1.2] | 2.7[1] | 2.8[1.2] | | 2.4[1.1] | 2.3[0.5] | 2.3[1] |
| Median [MIN, MAX] | 3.0 [0.0,5.0] | 3.0 [0.0,5.0] | 3.0 [0.0,5.0] | | 3[0,5] | 3[0,5] | 3[0,5] | | 3[0,5] | 3[1,4] | 3[0,5] | | 2[1,4] | 2[2,3] | 2[1,4] |
| **Fatigue During The Day** | | | | | | | | | | | | | | | |
| Mean [SD] | 2.9[1.1] | 2.7[1.2] | 2.9[1.1] | | 3[1] | 2.7[1.2] | 2.9[1.1] | | 2.9[1.2] | 2.7[1] | 2.9[1.1] | | 2.3[1.2] | 2.3[0.5] | 2.3[1.1] |
| Median [MIN, MAX] | 3.0 [0.0,5.0] | 3.0 [1.0,5.0] | 3.0 [0.0,5.0] | | 3[0,5] | 3[1,5] | 3[0,5] | | 3[0,5] | 3[1,4] | 3[0,5] | | 2[1,5] | 2[2,3] | 2[1,5] |
| **Reduced Productivity** | | | | | | | | | | | | | | | |
| Mean [SD] | 3.0[1.1] | 2.9[1.4] | 3.0[1.1] | | 3.1[1] | 2.9[1.5] | 3[1.1] | | 2.7[1.2] | 2.3[0.5] | 2.6[1.1] | | 2.7[1.2] | 2.3[0.5] | 2.6[1.1] |
| Median [MIN, MAX] | 3.0 [0.0,5.0] | 3.0 [0.0,5.0] | 3.0 [0.0,5.0] | | 3[0,5] | 4[0,5] | 3[0,5] | | 3[1,5] | 2[2,3] | 2[1,5] | | 3[1,5] | 2[2,3] | 2[1,5] |
| **Reduced Concentration** | | | | | | | | | | | | | | | |
| Mean [SD] | 2.9[1.2] | 2.9[1.3] | 2.9[1.2] | | 3[1.2] | 2.8[1.4] | 3[1.2] | | 2.7[1.2] | 3[1.2] | 2.8[1.2] | | 2.9[1.4] | 2.8[1] | 2.9[1.3] |
| Median [MIN, MAX] | 3.0 [0.0,5.0] | 3.0 [0.0,5.0] | 3.0 [0.0,5.0] | | 3[0,5] | 3[0,5] | 3[0,5] | | 3[0,5] | 3[1,5] | 3[0,5] | | 3[1,5] | 2.5[2,4] | 3[1,5] |
| **Frustated Restless Irrtable** | | | | | | | | | | | | | | | |
| Mean [SD] | 2.9[1.1] | 2.9[1.6] | 2.9[1.2] | | 3[1.1] | 2.8[1.6] | 2.9[1.2] | | 2.9[1.1] | 3.1[1.6] | 3[1.2] | | 2.6[1.1] | 2.8[1.5] | 2.6[1.2] |
| Median [MIN, MAX] | 3.0 [0.0,5.0] | 3.0 [0.0,5.0] | 3.0 [0.0,5.0] | | 3[1,5] | 3[0,5] | 3[0,5] | | 3[0,5] | 3[1,5] | 3[0,5] | | 3[1,5] | 2[2,5] | 2[1,5] |
| **Sad** | | | | | | | | | | | | | | | |
| Mean [SD] | 2.8[1.1] | 2.7[1.6] | 2.8[1.2] | | 2.8[1.1] | 2.6[1.5] | 2.8[1.2] | | 2.8[1.1] | 3.1[1.9] | 2.9[1.2] | | 2.4[1] | 2.8[1.5] | 2.4[1.1] |
| Median [MIN, MAX] | 3.0 [0.0,5.0] | 3.0 [0.0,5.0] | 3.0 [0.0,5.0] | | 3[0,5] | 3[0,5] | 3[0,5] | | 3[0,5] | 4[0,5] | 3[0,5] | | 2[1,5] | 2[2,5] | 2[1,5] |
| **Embrrassed** | | | | | | | | | | | | | | | |
| Mean [SD] | 2.6[1.1] | 2.4[1.5] | 2.6[1.2] | | 2.6[1.1] | 2.2[1.4] | 2.6[1.1] | | 2.6[1.2] | 3[1.7] | 2.7[1.3] | | 2.6[1.2] | 2[0] | 2.5[1.1] |
| Median [MIN, MAX] | 3.0 [0.0,5.0] | 3.0 [0.0,5.0] | 3.0 [0.0,5.0] | | 3[0,5] | 2[0,5] | 3[0,5] | | 3[0,5] | 4[0,5] | 3[0,5] | | 3[1,5] | 2[2,2] | 2[1,5] |
| **SNOT-22 score** | | | | *P* < 0.001 | | | | *P* < 0.001 | | | | *P* < 0.001 | | | |
| Mean [SD] | 65.3[16.4] | 56.4[20.0] | 63.7[17.4] | | 66.3[14.6] | 54.7[19.6] | 64.1[16.2] | | 63.9[17.9] | 62.9[18.3] | 63.7[17.8] | | 59.1[14.7] | 65[1.6] | 60.2[13.3] |
| Median [MIN, MAX] | 68.0 [3.0,95.0] | 69.0 [15.0,83.0] | 67.0 [3.0,95.0] | | 69[19,95] | 62[23,83] | 67[19,95] | | 67[21,93] | 71[25,78] | 67.5[21,93] | | 59[35,76] | 65[63,67] | 65[35,76] |
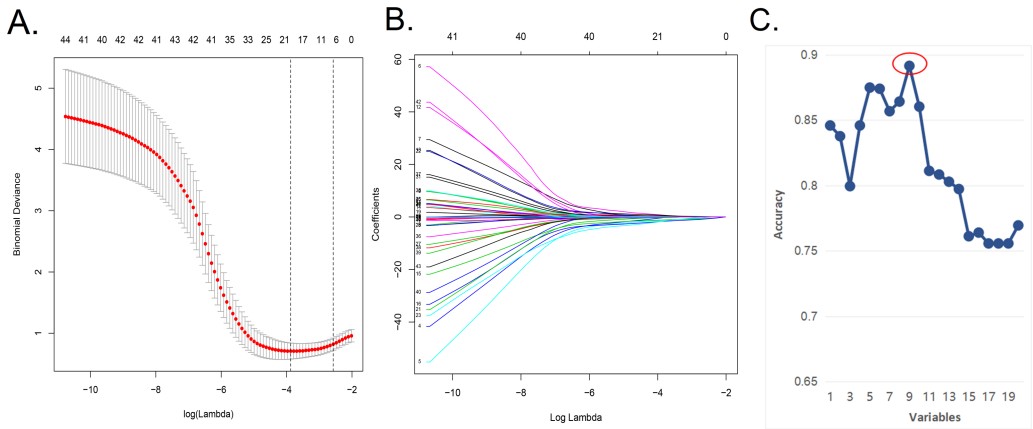

**Figure 1** **In the Lasso model, a five-fold cross-validation approach was used for the choice of optimal parameters.** (A) In the Lasso model, a five-fold cross-validation approach was used for the choice of optimal parameters.Using the partial likelihood anomaly curve and the log (lambda) plot, the vertical line was drawn at the optimal value to obtain the included feature factors. (B) The lambda curve generated a profile based on the log (lambda) sequence. Vertical lines were drawn at the values selected using the five-fold cross-validation method, with 20 characteristic factors being selected. (C) The algorithm of SVM-RFE support vector machine was used to further screen the 20 characteristic factors. Finally, a prediction model with 9 best features with an average 10-fold cross-validation score of 0.8914 was established.

**Table 2** **Rank the order of features in SVM-RFE method.**

| Features | Coefficients | Rank the order |
|---|---|---|
| Middle turbinate plasty | −4.75266432 | 1 |
| Nasosinusitis | −2.21413383 | 2 |
| Nasal surgery history | −2.3517654 | 3 |
| Seansonal allergy history | 3.03876115 | 4 |
| Age | −0.06223234 | 5 |
| SNOT22 Score | 0.03536382 | 6 |
| BMI | −0.17801399 | 7 |
| Smoke | −1.088097 | 8 |
| Follow up time | −0.35657536 | 9 |

validation queue, the overall sample, and the external dataset, respectively. In addition, the AUC of the prediction model were 0.920 in training set (the AUC curve is shown in Fig. 3B). For our research, we believe that increasing the sample size will be the main solution to overfitting. We have tested the out-of-group samples according to your suggestions and still obtain a high accuracy (Fig. 3B). Therefore, we believe that using larger sample size in future research will greatly decrease the risk of overfitting.

## Clinical application

The decision curve analysis of the efficacy prediction model of endoscopic NSD surgery is shown in Fig. 3C. The decision curve revealed that when the threshold probability of a patient and doctor is 20% and 100%, respectively, in the entire cohort, using this

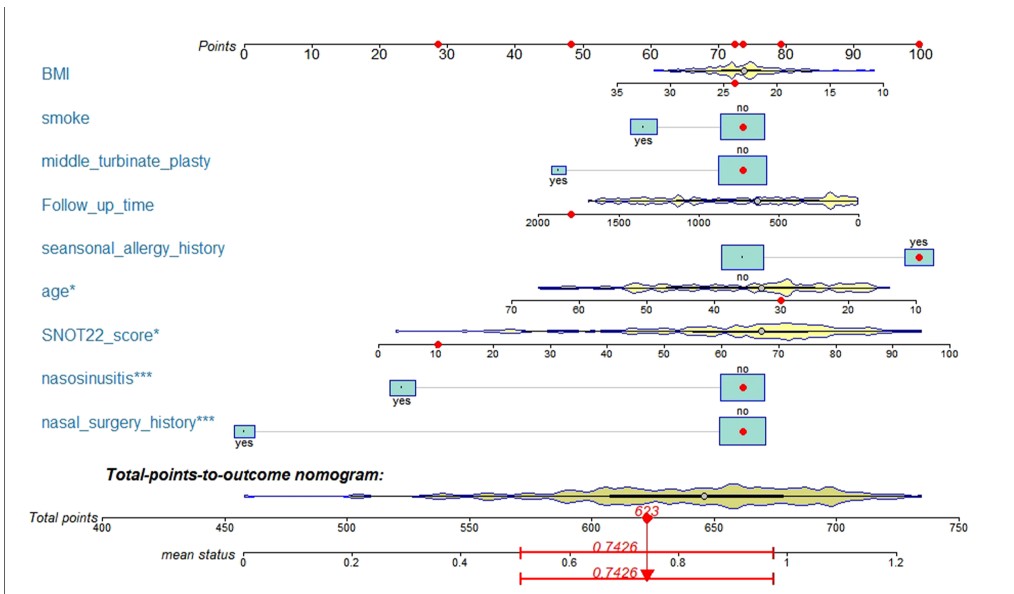

**Figure 2** **A nomogram model predicting the likelihood of benefit from surgery.** Note: nine factors including history of nasal surgery, preoperative SNOT-22 score, sinusitis, middle turbinate plasty, BMI, smoking, follow-up, advanced age, and seasonal allergies were included. * $p < 0.05$,*** $p < 0.005$.

**Table 3    Chart of prediction factors.**

| Variable | Prediction model | | |
| --- | --- | --- | --- |
| | $\beta$ | Odds ratio (95% CI) | *P*-value |
| SNOT22 score | 0.034 | 1.035(1.004–1.070) | 0.028 |
| Age | −0.059 | 0.943(0.895–0.988) | 0.019 |
| Smoke | −0.678 | 0.508(0.154–1.680) | 0.259 |
| Seansonal allergy history | 2.189 | 8.930(1.817–74.700) | 0.017 |
| Nasosinusitis | −2.003 | 0.135(0.034–0.460) | 0.002 |
| Follow up time | −0.001 | 0.999(0.998–1.000) | 0.043 |
| BMI | −0.065 | 0.937(0.780–1.110) | 0.471 |
| Nasal surgery history | −2.823 | 0.059(0.013–0.227) | 0.0000975 |
| Middle turbinate plasty | −1.875 | 0.153(0.017–1.800) | 0.104 |

**Notes.**
$\beta$ is the regression coefficient.

nomogram to predict the efficacy of endoscopic nasal septoplasty provide additional benefits as reported previously (*Wang et al., 2018*). Within this range, net benefit was comparable with several overlaps, on the basis of the prediction nomogram. Therefore, the decision curve showed that clinical decisions based on the nomogram prediction model yielded better returns. This means that the current predictive model can achieve better clinical practice through early planning of clinical interventions and better prediction of surgical outcomes, therefore ensuring administration of the most suitable treatment.

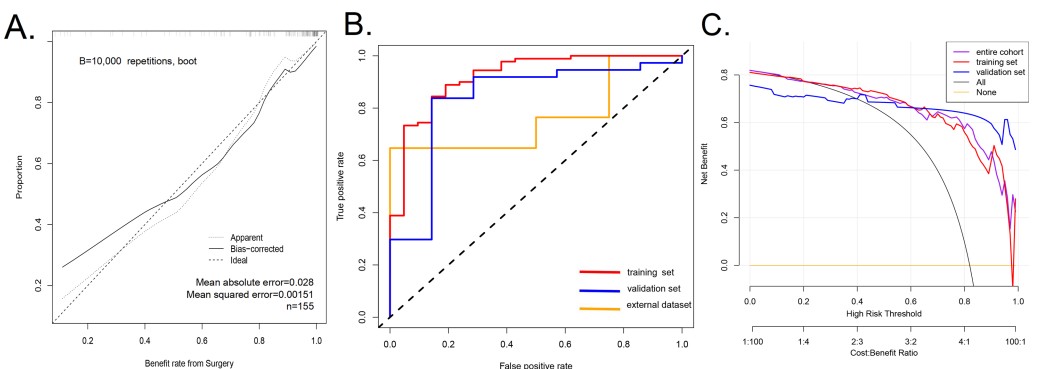

**Figure 3** **A calibration curve for the prediction model showing the benefits of endoscopic nasal septoplasty.** (A) A calibration curve for the prediction model showing the benefits of endoscopic nasal septoplasty. The diagonal dashed line represents a perfect prediction of an ideal model. The solid line indicates the predictive power of the predictive model, and an improved predictive ability was observed when it closely fitted with the dotted line. (B) The area under the curve (AUC) of the nomogram model indicates the probability of accurately predicting the likelihood of benefit from surgery in a randomly selected patient. The model exhibited good predictive power, with the AUC values of the training group (red), the test group (blue) and the external dataset (orange) recorded as 0.920, 0.834, and 0.765, respectively. (C) Decision curve used to estimate the surgical benefits. Decision analysis curves for the training, test, and overall groups are shown. The "None" line assumes that all patients failed to achieve the effect of surgery. The "All" line assumes that all patients achieved the effect of surgery.

# DISCUSSION

In this retrospective study, we established a continuum prediction model for surgical outcomes based on nine factors. The nomogram prediction model has good quantitative indicators and is suitable for prognosis and evaluation of clinical outcomes (*Wei et al., 2017*). To the best of our knowledge, this study is the first to apply a nomogram model to the evaluation of the efficacy of endoscopic NSD correction. A previous study validated the usefulness of the SNOT-22 score in patients with nasal septum deviation, but did not construct a predictive model based on the SNOT-22 score (*Pannu, Chadha & Kaur, 2009*). Early identification of patients with nasal septum deviation who may benefit from surgery will improve the management and optimal use of medical resources, and provide reasonable prediction of the surgical effects by doctors and patients. This will improve the doctor-patient relationship.

We created and validated a new tool for predicting the efficacy of endoscopic NSD correction. The tool uses nine readily available variables and has a high predictive accuracy. The incorporation of various characteristic factors into an easy-to-use nomogram facilitates individualized evaluation of the efficacy of endoscopic NSD correction. Our internal sampling showed the model's strong predictive ability. In addition, the high c-index and AUC index indicate that this predictive model can be widely and accurately used for the evaluation of therapeutic effect of nasal endoscopic deviation (*Wei et al., 2017*).

Studies that compare nasal patency before and after surgery (*Cantone et al., 2018*; *Hsu et al., 2017*) have been reported to have a higher risk of bias since treatment may improve the quality of life of patients, and be reflected in the SNOT-22 score. Therefore, in this study

we selected SNOT-22 score change as the primary outcome since it included quality of life for both general and specific diseases (*Van Egmond et al., 2019*). Moreover, assessment of postoperative quality of life in patients facilitates interdisciplinary research, and makes it possible to compare the depth of nasal septum to other interventions.

In this study, 81.9% of the patients attained satisfactory results after NSD surgery. In the risk factor analysis, history of nasal surgery, lower preoperative SNOT-22 score, combined sinusitis, combined with middle turbinate plasty, high BMI, smoking, longer follow-up, and advanced age were revealed to be the poor risk factors for NSD outcomes. Among these, SNOT-22 score, history of nasal surgery, sinusitis and age were the main factors affecting the efficacy of nasal septum surgery.

Our results suggest that preoperative SNOT-22 score is a good predictor of surgical outcomes. By definition, a SNOT-22 score change of 9 points after surgery is considered a good surgical outcome, a change that cannot be attained by patients with a lower preoperative SNOT-22 score. Therefore, the severity of a patient's condition has a bearing on the SNOT-22 score change, with more severe patients likely to record improvement in the quality of life after surgery. In addition, studies have shown that a preoperative SNOT-22 score of ≥ 20 can significantly improve the likelihood of surgical outcomes (*Hopkins et al., 2009*). The history of nasal surgery is a risk factor for the prognosis of surgical outcomes, since repeated nasal surgery often results into more complex lesions of local tissue, other than just a scar formation. We found that the probability of a nasal valve collapse during revision surgery was higher, and could be a likely cause of refractory nasal congestion (*Becker et al., 2008*), making efficacy of surgery in such patients worse.

Similar to most studies, we identified sinusitis as one of the risk factors for poor outcome in NSD surgery. NSD patients with CRS have been reported to have poor prognosis, longer treatment cycles, and higher consumption of medical resources (*Ahn et al., 2016*; *Bhattacharyya, 2005*). In addition, NSD also acts as a susceptibility factor for sinusitis by reducing nasal mucociliary activity, and inducing chronic inflammation and squamous metaplasia (*Ahn et al., 2016*; *Ji, Fu & Song, 2015*; *Kamani et al., 2014*; *Karatas et al., 2015*). Therefore, sinusitis treatment is a serious problem in the clinical treatment of NSD. Moreover, studies have shown that in patients with refractory sinusitis, no significant difference is observed between the efficacy of nasal septoplasty and conservative treatment methods (*Rudmik et al., 2011*). Our predictive model revealed that the middle turbinate plasty could affect the outcome of surgical treatment, making it a potential independent risk factor for sinusitis (*Javadrashid et al., 2014*). Concha bullosa, which is the aberrant pneumatization of the middle turbinate, affects the shape of the nasal septum leading to the occurrence of NSD (*Lee et al., 2008*). Incidence of Concha bullosa in patients with NSD has been shown to be higher in previous studies (*Yazici, 2019*). Patients with both nasal septum and Concha bullosa often display more severe histomorphological variation (*Tomblinson et al., 2016*). Subjective scoring for improvement in nasal symptoms and changes in short-term quality of life are often influenced by surgical comfort effects, hence should be performed early in the postoperative period. In our findings, we observed that patients with nasal septum surgery became less aggressive with prolonged postoperative time (*Bitzer,*

*2004*; *Bitzer, Dorning & Schwartz, 1996*; *Jessen, Ivarsson & Malm, 1989*; *Konstantinidis et al., 2005*).

In addition, over time, uncontrolled chronic sinusitis or other nasal structural lesions could affect the patient's subjective perception of nasal septoplasty, leading to a bias surgical evaluation (*Jessen, Ivarsson & Malm, 1989*; *Stewart, Robinson & Wilson, 1996*). Notably, the prognosis of smokers deteriorated faster than non-smokers. Studies have shown that after nasal surgery, smokers record longer recovery time due to slow wound healing process (*Cetiner, Cavusoglu & Duzer, 2017*). In addition, smoking also leads to a decrease in nasal mucociliary clearance (MCC) time (*Karaman & Tek, 2009*). This indicates that smoking could be a potential risk factor for poor prognosis in NSD surgery.

Recent studies have reported that among children and adolescents with severe symptoms, nasal septoplasty is a safe and effective procedure (*Fuller, Levesque & Lindsay, 2018*). However, patient satisfaction significantly declined with increase in age (*Habesoglu et al., 2015*). In addition, our multivariate analysis model identified advanced age as a risk factor for poor outcome after NSD (*Ahn et al., 2016*). Notably, the improvement of airway ventilation after NSD was more significant in young people, with the older patients not benefiting significantly after surgery (*Ratajczak et al., 2009*). Patients with NSD have poor nasal ventilation (*Glotov et al., 2017*), and obesity can aggravate the condition (*Ertugay et al., 2015*). Reports have shown that patients with a history of seasonal allergies in the nasal cavity have a better prognosis. This could be attributed to the poor adaptation of the nasal microenvironment to environmental changes in patients with severe NSD. However, the situation has been shown to be reversible though NSD surgery.

When considering surgery, clinical benefits, morbidity and associated complications, should be taken into consideration by patients and physicians. Therefore, an accurate prognosis would enhance doctors' assessment for the likelihood of patient's benefit and pre-operative communication with patients. In addition, this minimizes wastage of the costly medical resources. Therefore, in this study, we have developed an effective predictive model for the efficacy of NSD, which provides further theoretical guidance for clinical surgical treatment, and research of NSD. Despite the efficiency of our model, accurate determination of patient's benefit after surgery requires reasonable evaluation, and targeted interventions aimed at improving the postoperative efficacy of patients with NSD. To eliminate bias that could arise from differences in surgical proficiency between surgeons, all the operations were performed by an experienced surgeon. However, our study had the following limitations. First, the proportion of females in our sample was lower than males due to the higher risk of NSD in the latter. In addition, this study only focused on patients from China therefore does not represent patients with NSD in other countries and regions. Second, not all factors affecting the efficacy of endoscopic nasal septoplasty were included in the risk factor analysis. It is challenging to incorporate all objective factors in the statistical analysis. This is one of the dilemmas inherent in machine learning. We have attempted to obtain all the characteristic factors that we could reasonably access. Our results show that the factors obtained are representative, and they can effectively describe the prognosis of patients. Third, although our nomogram prediction model exhibited good prediction accuracy, its further optimization is required using more data. And SNOT-22

version is not yet fully content valid for NSD patients because other nasal symptoms common to NSD patients, such as epistaxis, which we added into our questionnaire, are not included in the current SNOT-22 architecture. Therefore, previous study suggests that further studies be performed to improve the SNOT score (*Leong & Webb, 2018*).

## CONCLUSIONS

In this study, we developed a robust predictive model that can be used by clinicians for the efficacy prediction of endoscopic NSD correction. Our model incorporates the history of nasal surgery, lower preoperative SNOT-22 score, combined sinusitis, combined with middle turbinate plasty, high BMI, smoking, longer follow-up, and advanced risk factors. We demonstrate that the use of this model for the prediction of the therapeutic effect of nasal endoscopic deviation, is effective and economical. In addition, ability of the clinicians to estimate individual risk ensures better communication with patients regarding the best treatment option. However, further research should be conducted to confirm the efficiency of this nomogram prediction model in predicting the efficacy and clinical utility of NSD correction.

### Funding

This work was supported by the Natural Science Foundation of Shanghai grants (14ZR14233800) and the Ninth Hospital Clinical Research Booster Program (JYL028). The funders had no role in study design, data collection and analysis, decision to publish, or preparation of the manuscript.

### Grant Disclosures

The following grant information was disclosed by the authors:
Natural Science Foundation of Shanghai grants: 14ZR14233800.
Ninth Hospital Clinical Research Booster Program: JYL028.

### Competing Interests

The authors declare there are no competing interests.

### Author Contributions

- Xue-ran Kang and Yi-sheng Chen conceived and designed the experiments, performed the experiments, analyzed the data, prepared figures and/or tables, authored or reviewed drafts of the paper, and approved the final draft.
- Bin Chen, Xiaojun Yan and Chenyan Jiang performed the experiments, authored or reviewed drafts of the paper, and approved the final draft.
- Bin Yi conceived and designed the experiments, prepared figures and/or tables, and approved the final draft.
- Shulun Wang analyzed the data, prepared figures and/or tables, and approved the final draft.

- Lixing Lu analyzed the data, authored or reviewed drafts of the paper, and approved the final draft.
- Runjie Shi conceived and designed the experiments, performed the experiments, prepared figures and/or tables, authored or reviewed drafts of the paper, and approved the final draft.

### Human Ethics

The following information was supplied relating to ethical approvals (i.e., approving body and any reference numbers):

This study was approved by the Institutional Ethics Review Board of Ninth People's Hospital affiliated to Shanghai Jiao Tong University Medical School (approval no. 2017-323-T243).

### Data Availability

The raw measurements are available in the Supplemental Files.

### Supplemental Information

Supplemental information for this article can be found online at http://dx.doi.org/10.7717/peerj.9890#supplemental-information.

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
