# Peer review of "A prediction modeling based on SNOT-22 score for endoscopic nasal septoplasty: a retrospective study"

_PeerJ, doi:10.7717/peerj.9890_

## Round 0.1 · original submission · Major Revisions

Dear authors,

Several experts have reviewed your paper and found scientific merit in your work. However, there are some issues that you should address in a revised version of the text.

Best regards,
Dr Palazón-Bru (academic editor for PeerJ)

Reviewer 1 ·

Basic reporting

Determination of the curative effect of nasal septoplasty is a good idea.
The study is well done.

Experimental design

OK

Validity of the findings

OK

Additional comments

Determination of the curative effect of nasal septoplasty is a good idea.
The study is well done.

Reviewer 2 ·

Basic reporting

Interesting article

Experimental design

Clear experimental design

Validity of the findings

Great validity of findings

Additional comments

Really interesting and well written article

Reviewer 3 ·

Basic reporting

I commend the authors for their extensive data analysis using SNOTT-22 score for the efficacy prediction of endoscopic nasal septum deviation correction. A normogram was created as a predictive model for the efficacy of nasal surgery.The manuscript is clearly written in professional English. The manuscript is structured well 3 descriptive figures and 2 tables. It is highlighted that the surgical outcome of nasal septal correction is affected by many factors including disease related factors, treatment related factors and patient related factors. The literature is well referenced and relevent.

Experimental design

A retrospective ethical approved study to develop a simple and effective tool that can be used by clinicians for the efficacy prediction of endoscopic nasal septoplasty. All the patients had to fulfill and inclusion and exclusion criteria. A total of 155 patients were included in the study with 109 males and 46 females with a mean age of 36.7 years. The methodology is well descibed with adequate details.

Validity of the findings

This novel study demonstrates that the use of normogram for nasal septal deviation surgery is both effective and economical.The limitation of this study was the gender was not equally represented and its limitation to mainland China only and a small sample study.

Additional comments

An extensive data analysis using SNOTT-22 score for the efficacy prediction of endoscopic nasal septum deviation correction. This study demonstrates that the use of normogram for nasal septal deviation surgery is both effective and economical.The limitation of this study was the gender was not equally represented and its limitation to mainland China only and a small sample study. It is a well written original primary research manuscript within the scope of the journal.

Reviewer 4 ·

Basic reporting

No comment

Experimental design

clinical implications should be better explain

Validity of the findings

Among limitations the authors reported that all the factors affecting the efficacy of nasal endoscopic septal deviation correction were not included in the risk factor analysis. This is a crucial pont. In addition, a morphologic classification about preoperative type of septal deviation should be added

Additional comments

Please add clinical and not only economical implications

Reviewer 5 ·

Basic reporting

- The authors proposed a prediction model for endoscopic nasal septoplasty. The idea is of interest and the use of English is acceptable.
- It is better to provide more background and literature reviews
- Article structure is ok, however some tables are displayed not too scientific. It is suggested to improve the display of their tables.
- SVM has been used in previous biomedical works such as PMID: 30822398 and PMID: 31055655, thus the authors are suggested to provide more references in this description.

Experimental design

- Why did the authors use SVM for feature selection, but logistic regression for classification? If the authors also use SVM for classification purpose, was the performance lower than logistic regression?
- When showing the patient characteristics, it is better to show p-value together to see the significant differences between two classes
- It is also better if the authors could report the training and validation cohort separately in Table 1.
- Measurement metrics or cross-validation had been used in different works e.g., PMID: 31277574 and PMID: 31380767. Therefore, it is important to refer more works to attract broader readership on this paper.

Validity of the findings

- It is important if the authors could validate their model on an external dataset.
- The authors should compare their performance results with the previous works on the same problem.
- How did the authors rank the order of features in their RFE method?
- From Fig. 2, it is easy to say that the model has overfitting (training performance was better than testing one). The authors should address this problem and discuss some solutions on it.
- In Fig. 2C, the legends stand in front of the curves, it makes the readers cannot see the curves.
- What are "All" and "None" lines in Fig. 2C?

Additional comments

No comment

---

## Round 0.2 · accepted · Accept

All the reviewers' concerns have been correctly addressed.

Reviewer 5 ·

Basic reporting

No comment

Experimental design

No comment

Validity of the findings

No comment

Additional comments

My previous comments have been addressed satisfactorily.